# A Comprehensive Review of Daridorexant, a Dual-Orexin Receptor Antagonist as New Approach for the Treatment of Insomnia

**DOI:** 10.3390/molecules27186041

**Published:** 2022-09-16

**Authors:** Wojciech Ziemichód, Karolina Grabowska, Antonina Kurowska, Grażyna Biała

**Affiliations:** Chair and Department of Pharmacology and Pharmacodynamics, Medical University of Lublin, 4a Chodźki Street, 20-093 Lublin, Poland

**Keywords:** daridorexant, insomnia, DORA, orexin system, orexin antagonists

## Abstract

Insomnia affects 4.4–4.8% of the world’s population, but because the effect of hypnotic drugs is limited and may cause harmful side-effects, scientists are turning their attention to developing drugs that act on the orexin system. Daridorexant, a selective dual-orexin receptor antagonist (DORA), has exhibited promising results in both animal and human studies. Its activity was evaluated based on the physiology-based pharmacodynamic and pharmacokinetic model. The use of daridorexant is considered safe, with no clinically significant side-effects including deprivation of next-morning residual effects. In this manuscript we conducted a comprehensive review of daridorexant including pharmacodynamics, animal and human research, pharmacokinetics and safety.

## 1. Introduction

Sleep is a physical and mental state characterized by altered consciousness, substantially inhibited sensory activity, reduced muscle activity, and, during rapid eye movement (REM) sleep, nearly complete inhibition of voluntary muscles [1]. Lack of sleep is known as insomnia, which has a negative impact on the functioning of the human body. Chronic dissatisfaction with sleep duration or quality, difficulty in falling asleep, numerous overnight awakenings with difficulties going back to sleep, and waking up earlier than desired are all symptoms of insomnia, but there is no universal theory to explain its pathophysiological cause [2].

Insomnia is thought to be the world’s most widespread sleeping disorder, and its prevalence may range from 4.4 to 4.8% depending on the diagnosis and testing methods [3,4]. According to U.S. research in 1994–1995, 23% of the population experienced symptoms of insomnia [3]. Similar research results were found in in Poland, where signs of insomnia were reported by 50.5% of respondents (58.9% women and 41.1% men) [5,6].

Numerous effective pharmaceutical remedies focus on different elements of recognized pathophysiologic mechanisms. For instance, benzodiazepine receptor agonists such as temazepam and zolpidem are frequently successful in treating insomnia by increasing the generalized inhibitory action of gamma aminobutyric acid (GABA). Doxepin, a tricyclic antidepressant, has demonstrated effectiveness for insomnia related to sleep initiation and maintenance; Trazodone is also commonly used although it has been not approved by the US Food and Drug Administration (FDA) for insomnia. The histaminergic arousal system is the target of the sedative action of these drugs [2,7].

The development of orexin receptor antagonists for treating insomnia has advanced, and the FDA has approved suvorexant for this use. These medications work to arouse brainstem/hypothalamic arousal areas by targeting the orexin system [2], which comprises the neuropeptides orexin A and orexin B and their two G-protein-coupled receptors—(GPCRs): orexin receptor 1 (OX1r) and orexin receptor 2 (OX2r) [8]. Initially, orexins were described as peptides that regulate eating behavior. However, further evaluation indicated that they are involved in a number of functions from modulation of autonomic functions to higher cognitive function including reward seeking, attention, behavior, mood, and cognition [8]. Interestingly, orexin signaling is also involved in mood disorders which are associated both with low hedonic tension or anhedonia, which includes depression, anxiety disorders, attention deficit hyperactivity disorder, and addiction [8]. As orexin neuropeptides are expressed in small neuronal populations in the hypothalamus, their activity is the highest during periods of activeness and wakefulness but non-active during sleep [9]. It is worth noting that orexin peptides promote neuronal populations including the histaminergic neurons of the tuberomammillary nucleus (expressing mainly OX2r), locus coerulus noradrenergic neurons (expressing mainly OX1r), serotoninergic neurons of the dorsal raphe and dopaminergic neurons of the ventral tegmental area (both expressing OX1r and OX2r) as well as the basal forebrain and the pedunculopontine [10]. Additionally, the laterodorsal tegmental nuclei of cholinergic neurons expresses both OX1r and OX2r [10]. The inhibition of both OX1r and OX2r reduces the activity of wake-promoting neurons and the wide-spread inhibition of neuronal pathways [10]. This new approach to the treatment of insomnia assumes the use of a dual orexin antagonist (DORA) such as daridorexant, the comprehensive characteristics of which are presented in this manuscript.

## 2. Pharmacodynamics of Daridorexant as a DORA Agent

Daridorexant, also known as nemorexant, is a novel benzimidazole derivative with the approved UPAC name [(2*S*)-2-(5-chloro-4-methyl-1*H*-benzimidazol-2-yl)-2-methylpyrrolidin-1-yl]-[5-methoxy-2-(triazol-2-yl)phenyl]methanone. It is a dual-orexin antagonist (DORA), and its structure is presented in Figure 1. The drug was identified as a sleep-promoting agent with appropriate duration of action with no next-morning residual activity. The drug was evaluated based on physiology-based pharmacokinetic (PK) and pharmacodynamic (PD) modeling [9]. Daridorexant not only decreases wakefulness but also promotes sleep while preserving the sleep architecture [11,12]. DORAs are considered to be a new alternative for well-known positive allosteric GABA-A receptor modulators [12]. Because it was established in intracellular Ca^2+^ release assays, the drug acts as a competitive orthosteric antagonist [12]. The Kb values were established as 1.1 nM in a rat, 0.3 nM in dog and 0.5 nM in human models with OX1r and 1.7, 0.7 and 0.8 nM for OX2r for rat, dog and human respectively (Kb values were derived from IC50 values using the generalized Cheng–Prusoff equation and their geometric mean) [10,12]. Of particular note, Treiber et al. (2017) established that the compound was selective against OX1r and OX2r in a panel screen of more than 130 established central and peripheral pharmacological targets [12]. The selectivity and mechanism of action of daridorexant as an orthosteric antagonist are advantages over typical hypnotic drugs because it does not cause typical side effects. It was established that the higher activity of daridorexant was associated with high orexin neuron activity. During sleep, orexin release is ramped up to counteract increasing insomnia, sleep deprivation as well as activity phase, whereas it should have limited effects following a good night’s sleep [10]. Of further note, Treiber et al. evaluated daridorexant brain penetration in rats. The penetration was established 3 h after oral administration of daridorexant in doses of 30 and 100 mg/kg. After administration of the 30 mg/kg dose, the concentration of daridorexant reached 665 nM, whereas total brain concentration increased in a more than dose-dependent manner at 100 mg/kg reaching 2247–12,000 nM [12].

## 3. Animal and Human Studies on the Hypnotic Activity of Daridorexant

The hypnotic activity of daridorexant was confirmed in a number of studies conducted on animals and humans. During evaluation, not only was hypnotic activity evaluated but also its influence on the sleep architecture and safety of use, which is presented later the manuscript.

During their evaluation, Treiber et al. stated that the dose response was established at 10, 30 and 100 mg/kg. That experiment was conducted on male Wistar rats with implanted telemetric transmitters that allowed for continuous EEG/EMG- based evaluation of sleep/wake cycles. Thus, it was possible to determinate the length of the REM and non-REM phases. Oral administration of daridorexant in a dose-dependent manner influenced sleep-wake parameters and decreased active wake time by 22% compared to the rats treated with vehicle. Additionally, the drug prolonged the time spent in both REM and non-REM sleep by 84 and 29%, respectively (10 and 26 min, respectively). However, the time from latency to persistent non-REM sleep (LPS) decreased by 59% from 44 to 18 min; for the REM sleep phase it was 58% (30-71 min). As it was reported, the lowest dose that had significant effects was 30 mg/kg [12]. 

Muehlan et al. conducted double-blind, randomized, placebo-controlled studies of a single oral dose on 40 healthy males. The patients were divided into groups and received a dose of 5, 25, 50, 100 or 200 mg of daridorexant in the morning under fasting conditions. The effects at the 5 mg dose were hardly detectable, but doses of 25 mg or higher had a potent influence on the central nervous system. The subjects were observed to have reduced vigilance, attention, vasomotor symptoms, coordination and postural stability as indicated by a decreased saccadic peak velocity (SPV). Adaptive tracking performance and increased body sway were also observed. All of these effects occurred 1 h after administration, whereas a maximum effect occurred around 1.5 h following the intake of doses of up to 100 mg. After the administration of 200 mg, a maximum effect was observed after 2 h. The effects of daridorexant at 25 and 50 mg doeses returned to baseline within 3–6 and 6–8 h, respectively, whereas after the 100 and 200 mg does, the return to baseline for most variables was 8–10 h. It is worth mentioning that in the group that received the 200 mg dose, the SPV effects and visual analog scale (VAS) subjective alertness did not completely return to baseline by the end of the 10 h observation period [13].

In 2019 Muehlan et al. conducted further evaluation of the dose-dependent pharmacodynamic effect of daridorexant on a group of 51 patients. It was reported that there were no pharmacodynamic effects on day 1 or 5 after administration of 10 mg in the morning. However, in the 25 and 75 mg groups, all pharmacodynamic variables showed effects including the motor and cognitive functions of the central nervous system. Additionally, the drug reduced vigilance, attention, visual-motor coordination as well as postural stability. The onset of objective pharmacodynamic effects was within 1 h after administration; maximum effect was observed around 2 h; and the return to baseline occurred within 4–10 h. On day 1 in both the 25 and 75 mg dose groups, decreased SPV had a maximum mean reduction of 57 for the 25 mg and 87 for 75 mg doses. On day 5, maximum mean reduction of SPV was observed 1 h after dosage with a similar magnitude as on day 1. The return to baseline was observed approximately 10 h after dosing. Decreased adaptive tracking performance was also observed in a dose-dependent manner after administration of 25 and 75 mg. The maximum mean reduction on day 1 was estimated as 4.6 and 16.8% on day 5. Increased body sway was observed with maximum mean values observed on day 5 of 154 and 476 mm for 25 and 75 mg, respectively. Additionally, subjective alertness decreased dose-dependently, with a maximum mean reduction of 7.4 and 10.4% for 25 and 75 mg, respectively. Using the Karolinska Sleepiness Scale (KSS), small mean maximum effects of 1.5–3 were observed on day 1 and day 5. At 75 mg, the small mean maximum increased from baseline in the range 0.1–0.5 in all the four components of the visual analog scale (VAS). There were no relevant differences in the magnitude of effects on day 1 or day 5 to any of the PD parameters. 

Worth mentioning is that the next-day PD effects of daridorexant were observed (SPV, adaptive tracking, body sway, VAS Bond and Lader) and assessed 8 h. post dosage on the morning of day 2 and day 8. No relevant mean changes from baseline were observed compared to placebo [14].

The dose-response of daridorexant on waking after sleep onset was also investigated in the evaluation conducted by Zammit et al. (2020) on a group of 58 participants diagnosed with insomnia. They were randomly allocated according to Latin square design into 5 groups. In each group, patients received 5, 10, 25 or 50 mg of daridorexant or a placebo. Every 5 treatment periods consisted of 2 treatment nights, followed by a 5- to 12-day period of washout. The main efficiency endpoints were the absolute change from baseline in wakefulness after sleep onset (WASO) (primary endpoint) and LPS (secondary endpoint) for days 1 and 2 in each period. Both WASO and LPS were dose-dependently reduced from baseline to days 1 and 2 after administration of daridorexant. The most statistically meaningful reduction was for doses higher or equal to 10 mg compared to placebo, which was in accordance with the results obtained by Muehlan et al. In this evaluation, WASO values was reduced by −32.0, −45, and −61.4 min. for 10, 25 and 50 mg, respectively, whereas LPS values were reduced by −44.9, −43.8 and −45.4 min for 10 25 and 50 mg respectively [15]. According to research conducted by Dauvilliers et al. (2020) on a group of people with insomnia, daridorexant caused a dose-dependent reduction in WASO. The evaluation was conducted with adults under the age of 64 who were randomized (1:1:1:1:1:1) to receive a placebo, daridorexant in a dose of 5, 10, 25 or 50 mg or 10 mg of zolpidem for 30 days. As a result, the significant dose–response relationship at 4 weeks had noticeable WASO (*p* = 0.045), whereas it was not relevant for sLSO (time to fall asleep) (*p* = 0.107). However, the subjective total sleep time (sTST) increased in a dose dependent manner (*p* = 0.006). It is worth mentioning, that the absolute change from baseline to day 30 was similar between placebo and daridorexant but smaller compared to zolpidem. In all of the groups, sleep quality was judged to be better at week 4 than week 2 and there were no differences in sleep parameters in both male and female groups. What is important, is that the evaluated insomnia severity index was not dose dependent [9]. 

Subsequent confirmation that daridorexant improves sleep and daytime functioning in people with insomnia was presented in *The Lancet* in 2022. Mignot et al. conducted two multicentre, randomized, double-blind, placebo-controlled phase 3 trials on 930 participants in 156 cities in 17 countries. The subjects were randomly assigned to receive for 3 months 50 or 25 mg daridorexant or placebo (study 1) or 25 or 10 mg or placebo (study 2) every evening. The primary evaluated endpoints were changed from baseline in WASO and LPS, which were measured by polysomnography at months 1 and 3. Secondary endpoints were a change in the value of the baseline of sTST and the sleepiness domain score on the insomnia daytime symptoms and impact questionnaire (IDSIQ) at months 1 and 3. As a results, both WASO and LPS were significantly reduced in a group who received daridorexant at the 25 and 50 mg doses compared to placebo at months 1 and 3. All participants also reported improved total sleep time as well as IDSIQ sleepiness at months 1 and 3. In study 2, WASO was also reduced at the 25 mg dose compared with the placebo group at months 1 and 3. However, no significant changes were reported in LPS at months 1 and 3. What is important is that there was no significant difference between the placebo group and the 10 mg daridorexant group. This finding corresponded with the research conducted by Muehlan et al. (2018), who reported that effects of daridorexant at the 5 mg dose were barely detectable [16].

## 4. Pharmacokinetics of Daridorexant

Evaluating the duration of action of a new drug during research and introduction into treatment is critical. The pharmacokinetic properties of daridorexant are not clinically affected by age, sex, race, body size, or mild-to-severe renal impairment (Cockcroft-Gault 30 mL/min, not on dialysis) [17,18].

Daridorexant had proportional plasma exposure at doses of 25–50 mg. The drug’s pharmacokinetic profile was similar after multiple-dose and single-dose administration with no accumulation and with a time to peak plasma concentrations of 1–2 h (tmax) [19]. A high-fat/high-calorie meal delayed the t max of daridorexant by 1.3 h and reduced the maximum concentration of the drug (Cmax) by 16%, but had no effect on total exposure (area under the curve (AUC)). The drug had a 31 L volume of distribution and was 99.7% bound to plasma proteins. The terminal half-life (t 1/2) of daridorexant was 8 h in all studies [13,17].

Daridorexant is extensively metabolized, primarily by CYP3A4 (89%), and mostly excreted via feces (57%) and urine (28%) as metabolites [17]. The metabolite formulas presented in Figure 2a are a primary alcohol formed by hydroxylation of the benzimidazole ring’s methyl group. This reaction is primarily catalyzed by CYP 3A4. 

Daridorexant 50 mg can be co-administered with the histamine 2 receptor inhibitor famotidine without changing the dosage [21]. The combination of daridorexant 50 mg and the selective serotonin reuptake inhibitor (SSRI) to treat depression and anxiety did not result in clinically significant changes in the pharmacokinetic parameters [22].

The coadministration of daridorexant with the lipid-lowering rosuvastatin, a specific 3-hydroxy-3-methylglutaryl-coenzyme reductase inhibitor, had no effect on the latter’s pharmacokinetics, suggesting that daridorexant could be safely co-administered with breast-cancer resistant protein (BCRP) substrates without dosage adjustments [23]. 

Coadministration of 25 mg daridorexant with the mild CYP3A4 inhibitor diltiazem increased daridorexant AUC by 240%, and coadministration with the strong CYP3A4 inhibitor itraconazole is expected to increase AUC by more than 400%, according to physiologically based pharmacokinetic modeling [24]. Coadministration with the moderate CYP3A4 inducer efavirenz, on the other hand, reduced daridorexant AUC by 35% [21], and coadministration with the strong CYP3A4 inducer rifampin is expected to reduce AUC by more than 50% [24]. On this basis, the maximum recommended dose of daridorexant is 25 mg when used in conjunction with a moderate CYP3A4 inhibitor, but use in conjunction with a powerful CYP3A4 inhibitor or a modest or strong CYP3A4 inducer is not advised [17]. The presence of moderate Child–Pugh B but not mild Child-Pugh A liver impairment has been shown to prolong the t 1/2 of daridorexant, so the maximum recommended dosage in patients with Child-Pugh B impairment is 25 mg no more than once per night [17,19]. The effects of severe hepatic impairment (Child-Pugh C) on the drug’s pharmacokinetics have not been studied, and daridorexant should not be used in this patient population [17].

Co-administration with alcohol increased the daridorexant tmax and was related to the additive effects on motor coordination performance. As a result, patients should avoid drinking alcohol while taking daridorexant [17,25], and when given with other CNS depressants, it should be used with caution, and dosage adjustments for either or both drugs should be considered.

## 5. Safety of Daridorexant and Its Abuse Potential

As every drug has its own side effects, it is understandable that there should be concerns, especially concerning a hypnotic drug. To help people overcome their worries scientists evaluated the safety of daridorexant as well as its abuse potential, which has been described in a number of studies.

Daridorexant is metabolized by CYP3A4, an essential enzyme, so it is important to be aware of any potentially negative effects when taking it either with an inhibitor or an activator of this subclass of enzyme.

According to physiologically based pharmacokinetic modeling, coadministration of 25 mg with the mild CYP3A4 inhibitor diltiazem enhanced the daridorexant AUC by 240%, and coadministration with the potent CYP3A4 inhibitor itraconazole is predicted to increase it by more than 400% [24]. In contrast, the moderate CYP3A4 inducer efavirenz reduced the AUC by 35% [21], and the strong CYP3A4 inducer rifampin is anticipated to reduce it by more than 50% [24]. In light of this, it is not advisable to use daridorexant in combination with a severe CYP3A4 inhibitor or a moderate or strong CYP3A4 inducer. The highest recommended starting dose is 25 mg when used with a moderate CYP3A4 inhibitor.

The use of hypnotic drugs by patients with breathing disorders can be controversial, especially since previous hypnotics such as the benzodiazepines and barbiturates caused respiratory depression. Boof et al. conducted studies of the effect of daridorexant on patients with obstructive sleep apnea (OSA). According to scientists, the use of daridorexant administered at the highest dose of 50 mg is safe [26]. The influence of the drug on nighttime respiration-related variables and sleep characteristics in patients with mild to moderate OSA were investigated in a randomized, double-blind, placebo-controlled crossover study. During the trial, the influence of daridorexant on OSA severity—the number and duration of apneas and hypopneas were monitored. Measurements were also taken of the mean and lowest nocturnal SpO_2_, sleep duration each hour of polysomnography recording as well as the number, mean and longest duration of awakenings [26]. As the scientists reported, after multiple administrations of daridorexant, the number of respiratory events increased compared to placebo, but they attributed it to prolonged total sleep time. Worth highlighting is that there was no coincidence between the longest duration of apneas and hypopneas and the lowest SpO_2_. It was also noticed that the number of awakenings was comparable in both the daridorexant and placebo groups. However, daridorexant shortened the longest duration by 16.2 min. In conclusion, scientists claimed that the evaluation of a various of indices of OSA severity and sleep suggested that daridorexant was safe at the 50 mg dose for patients with mild to moderate OSA [26]. 

Schilling et al. further evaluated the safety of daridorexant concerning cardiac repolarization. In s double-blind, randomized, placebo-controlled and four-period crossover study, 36 healthy patients received 50 or 200 mg of daridorexant, or 400 mg of moxifloxacin (included as a positive control to detect a relevant QT prolongation) or a placebo. All of medications were administered at bedtime. The fundamental aim of the study was to evaluate the effect of single doses of daridorexant on QTcF. All patients went through four treatment periods separated by a wash-out period of 7–10 days. Data were collected from triplicate ECGs from a Holter recording at baseline and after 24 h after dosing at times matching those for the pharmacokinetic samples. As a result, moxifloxacin exhibited the expected magnitude of QTcF prolongation with a maximum delta QTcF of 13.6 ms at 8 h. However, by the time point analysis of delta QTcF, both daridorexant and the placebo exhibited a magnitude and time course. The mean of delta QTcF remained below 10 ms at all time points. Interestingly enough, at all of the time-points and both doses of daridorexant (50 mg or 200 mg), the mean values of delta QTcF ranged from 1.92 ms to 3.77 ms. Thus, it indicated no relevant correlation between the use of daridorexant and cardiac repolarization [27].

The Swiss scientists Muehlan et al. examined the impact of daridorexant on driving performance using a driving simulator. In the examination, 67 patients were randomized in a placebo- and active- controlled, 4-way cross-over study. Every patient received 50 or 100 mg of daridorexant, 7.5 mg of zopiclone or a placebo in separate treatment phases of four days. Simulated driving performance was conducted after initial (day 2) and repeated dosing (day 5) 9 h post-dose. The main outcome was the deviation of the lateral position (SDLP). As it was reported on both days, SDLP values in the zopiclone treatment group increased significantly compared to the placebo: 4.75 cm on day 2, and 2.37 cm on day 5. In the daridorexant group on day 2, the placebo-corrected mean SDLP increased by 2.19 cm for patients treated with the 50 mg dose, and 4.43 cm for those treated with 100 mg. Importantly, the values of the SDLP for both daridorexant doses were significantly below the prespecified threshold of impairment on day 5. (2.6 cm), which was not statistically different from the placebo. In the zopiclone group, significantly more subjects exhibited impaired driving (62.5%) compared with improved (5.4%) driving on day 2 as well as on day 5 (50.0% impaired driving vs. 12.5% improved). In the group treated with daridorexant 50 mg, on day 2 a difference in the number of impaired vs. improved subjects was estimated as 43.6 and 14.5%, respectively. However, in the 100 mg group, the percentage of patients with impaired driving was 65.5% whereas 6.9% improved. Worth noticing is the fact, that on day 5 daridorexant treatment was statistically not different from placebo (25.9% impaired vs. 27.6% improved, and 24.1% impaired vs. 12% improved performance with 50 and 100 mg, respectively). As a result, for patients without insomnia daridorexant impaired simulated driving after the initial dose, however, it did not after repeated dosing [28].

During all the clinical trials described in this article, non-hazardous side effects were observed. For example, during clinical studies by Mignot et al. (2022), the frequency of side effects was comparable between both studies. The most common reported side effect was nasopharyngitis—also reported in the evaluation conducted by Zammit et al.—and headache—which was also reported in Muehlan et al. [28], Shilling et al. [27], and Zammit et al. [15]. Importantly, there was no evidence of dose dependency between the side effects and the dose [16]. Additionally, somnolence and fatigue were reported, especially at the beginning of the administration [28] as was gait disturbance [15].

Because abuse had been reported for the previous DORA agents survorexant and lemborexant, Ufer et al. (2022) examined the abusive potential of daridorexant. The trial was randomized, double-blind, double-dummied, placebo-, active- controlled with a six-period crossover. In the evaluation, 63 patients were recruited. In each study period, a single, oral, morning dose of daridorexant (50, 100 or 150 mg), placebo, zolpidem (30 mg) or suvorexant (150 mg) was administrated. The primary pharmacodynamic endpoint was the maximum effect (Emax) of the drug-like VAS, which was assessed over 24 h. Additionally, several secondary subjective and objective endpoints were assessed. The validity of the study was confirmed because zolpidem and suvorexant were higher than the placebo after applying a predefined 15-point validity margin. Although the abuse potential was estimated to be lower compared to suvorexant or zolpidem at 50 mg, the range was similar at dose 100 and 150 mg. The Drug-like VAS Emax of daridorexant at 50 mg was estimated at a mean value (MV) of 73.2, whereas the VAS Emax for suvorexant was 80.7 (MV) and 79.9 (MV) for zolpidem. The values of VAS Emax for daridorexant at 100 mg was 79.1 and 81.3 (MV) for 150 mg. As a conclusion, the scientists indicated that although daridorexant exhibited dose-related drug-like behavior among recreational sedative drugs, it had lower effects at the highest phase-3 dose. However, it had similar effects at higher doses when compared to supratherapeutic doses of suvorexant and zolpidem. Simultaneously, scientist implied that daridorexant is safe and the pharmacokinetic were consistent with earlier trials [29].

## 6. Conclusions

Daridorexant, a new selective DORA agent, acts as competitive orthosteric antagonists and exhibits good brain penetration. Importantly, it promotes sleep with preserved sleep-architecture, and its action leaves no next-morning side effects, which sets it apart from typical hypnotic agents such as benzodiazepines. The activity of daridorexant was confirmed in a number of both animal and human trials that allowed evaluating the impact of daridorexant on the duration of REM and non-REM phase, appropriate dosage, pharmacokinetics and the influence of daridorexant on daily functioning. All of these studies led to a better understanding of its mechanism of action and safety. It was established that the drug is safe for patients with mild to moderate OSA at a dose of 50 mg, and it did not relevantly influence cardiac repolarization. Additionally, the influence of daridorexant on driving showed that it was safer than other hypnotic drugs. Therefore, it seems that daridorexant can become a hypnotic alternative to improve the life of patients suffering from insomnia.

## Figures and Tables

**Figure 1 molecules-27-06041-f001:**
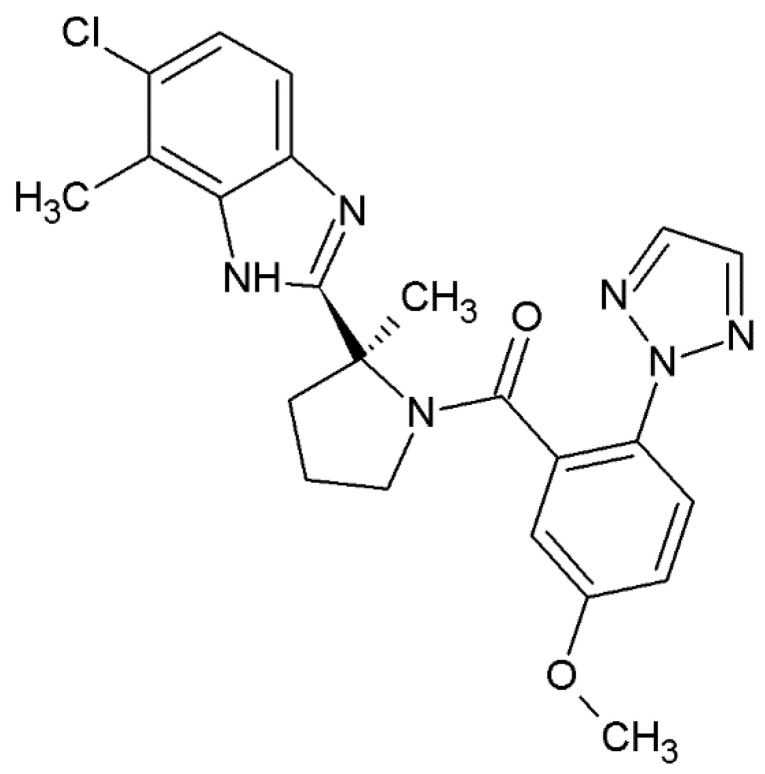
C structure of daridorexant.

**Figure 2 molecules-27-06041-f002:**
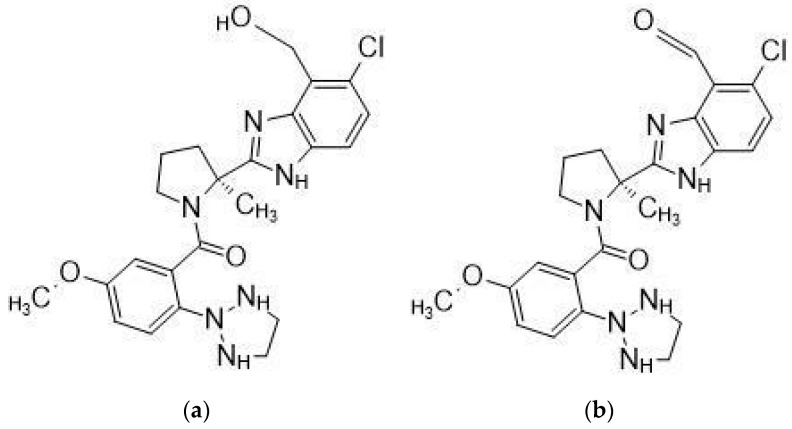
Major daridorexant metabolites. (**a**) is the starting point for (**b**), which is formed by oxidizing the primary alcohol to the corresponding aldehyde, then (**c**) is formed by an oxidative rearrangement of metabolite (**d**) [20].

## Data Availability

Not applicable.

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
