# Peer review of "A Comprehensive Review of Daridorexant, a Dual-Orexin Receptor Antagonist as New Approach for the Treatment of Insomnia"

_molecules, 2022, doi:10.3390/molecules27186041_

Round 1

Reviewer 1 Report

Review Article titled: A Comprehensive Review of Daridorexant-Dual Orexin Recep- 2 tor Antagonist.

Authors : Ziemichód et al.,

Daridorexant (Quviviq) approved novel dual orexin receptor antagonist indicated for the treatment of adult patients with insomnia characterized by difficulties with sleep onset and/or maintenance.  Daridorexant exhibits its effect through blocking the binding of wake-promoting neuropeptides orexin A and orexin B to the receptors OX1R and OXR2. The antagonism at these receptors is thought to suppress overactive wakefulness.

The authors explained the review in the following subtitles: Pharmacodynamics of daridorexant as a DORA agent, Animal and human studies on the hypnotic activity of daridorexant, Pharmacokinetics of Daridorexant, and Safety of daridorexant and its abuse potential.

The topic is very interesting as the drug development based on orexin system getting more attracted by pharmaceutical companies and the treatment not only limited to insomnia but also applied to prevent addiction.

It is well written manuscript.

There are some minor comments:

1.       Fig. 2 in two pages. It makes the feeling that two figures. There is no labels for the figures.

2.       Correct the spacing of Line 93, Line 136, Line 252, and line 285.

3.       Line 264-273 make into one paragraph.

4.       Some references are not given the numbers but gave years: Line 84, line 108, line 120,line 166, Line 179, line 209, line 350, line 358.

Author Response

-Dear Editor, thank You very much for Your review. We are honored to hear that from You.

- Fig. 2 in two pages. It makes the feeling that two figures. There is no labels for the figures.

-Thank You for this remark. The figures were change according to Your indications. We also explained the origin of the figures.

  1. Correct the spacing of Line 93, Line 136, Line 252, and line 285.

-Thank You for pointing this out. The spacing was corrected.

  1. Line 264-273 make into one paragraph.

-Thank You for Your suggestion. The lines were converted into one paragraph according to Your suggestion.

  1. Some references are not given the numbers but gave years: Line 84, line 108, line 120,line 166, Line 179, line 209, line 350, line 358.

- The references are added at the end of the paragraph. According to Yours indications we deleted years written next to the name of the authors in pointed lines. Thank You for pointing this out. I am convinced that this change makes the manuscript more preferable.

Reviewer 2 Report

1. The title of the paper should be a little bit change as the Authors focused mainly on daridorexant properties and activities in insomnia. Otherwise the title suggests that this drug is characterized in every possible field.

2. Although the Authors provided the information about antagonistic behavior of the drug by giving Kb values, it would be nice to know the IC50 and Emax vaules towards orexin receptors.

3. In the section 3, the Authors stated "Treiber and coworkers (2017) already cited, during their evaluation stated that the dose response was estabilished at...." Please provide additional information on the way of drug administration.

4. Since the drug is metabolized by one of the crucial enzyme CYP3A4, the Authors should also pay attention on possible side effects induced by this drug when combined either with inhibitor or activator of this subclass of enzyme. In fact, some information have already been given by the Authors in the section Pharmacokinetics, however, similar should be provided in the section regarding daridorexant safety profile

5. Please check carefully the style. Also, some typos can be found, e.g. tamental instead of tegmental,

Author Response

-Daridorexant (Quviviq) approved novel dual orexin receptor antagonist indicated for the treatment of adult patients with insomnia characterized by difficulties with sleep onset and/or maintenance.  Daridorexant exhibits its effect through blocking the binding of wake-promoting neuropeptides orexin A and orexin B to the receptors OX1R and OXR2. The antagonism at these receptors is thought to suppress overactive wakefulness.

The authors explained the review in the following subtitles: Pharmacodynamics of daridorexant as a DORA agent, Animal and human studies on the hypnotic activity of daridorexant, Pharmacokinetics of Daridorexant, and Safety of daridorexant and its abuse potential.

The topic is very interesting as the drug development based on orexin system getting more attracted by pharmaceutical companies and the treatment not only limited to insomnia but also applied to prevent addiction.

It is well written manuscript.

-Dear Editor, thank You very much for Your review. We are honored to hear that from You.

- Fig. 2 in two pages. It makes the feeling that two figures. There is no labels for the figures.

-Thank You for this remark. The figures were change according to Your indications. We also explained the origin of the figures.

  1. Correct the spacing of Line 93, Line 136, Line 252, and line 285.

-Thank You for pointing this out. The spacing was corrected.

  1. Line 264-273 make into one paragraph.

-Thank You for Your suggestion. The lines were converted into one paragraph according to Your suggestion.

  1. Some references are not given the numbers but gave years: Line 84, line 108, line 120,line 166, Line 179, line 209, line 350, line 358.

- The references are added at the end of the paragraph. According to Yours indications we deleted years written next to the name of the authors in pointed lines. Thank You for pointing this out. I am convinced that this change makes the manuscript more preferable.

Reviewer 2

-The title of the paper should be a little bit change as the Authors focused mainly on daridorexant properties and activities in insomnia. Otherwise the title suggests that this drug is characterized in every possible field.

- Thank You for bringing to our notice how misleading the previous title was. The title was changed according to Your indications.

  1. Although the Authors provided the information about antagonistic behavior of the drug by giving Kb values, it would be nice to know the IC50 and Emax vaules towards orexin receptors.

-Thank You for pointing this out. We added information that Kb values derived from IC50 using the generalized Cheng-Prusoff equation as it is described in cited manuscript (page 2).

  1. In the section 3, the Authors stated "Treiber and coworkers (2017) already cited, during their evaluation stated that the dose response was estabilished at...." Please provide additional information on the way of drug administration.

-Thank You for bringing to our notice how relevant this information is. The information was added according to Your indication (page 3).

  1. Since the drug is metabolized by one of the crucial enzyme CYP3A4, the Authors should also pay attention on possible side effects induced by this drug when combined either with inhibitor or activator of this subclass of enzyme. In fact, some information have already been given by the Authors in the section Pharmacokinetics, however, similar should be provided in the section regarding daridorexant safety profile

-Thank You for Your suggestion. The information was added in indicated section regarding safety profile (section “safety of daridorexant”, page 8).

  1. Please check carefully the style. Also, some typos can be found, e.g. tamental instead of tegmental,

- We appreciate Your remark. The manuscript was carefully read and the mistakes were corrected.
